# Longitudinal SARS-CoV-2 Seroprevalence among Employees in Outpatient Care Services in Hamburg

**DOI:** 10.3390/ijerph20085599

**Published:** 2023-04-20

**Authors:** Anja Schablon, Volker Harth, Claudia Terschüren, Olaf Kleinmüller, Claudia Wohlert, Claudia Schnabel, Thomas Theo Brehm, Julian Schulze zur Wiesch, Jan Felix Kersten, Albert Nienhaus

**Affiliations:** 1Competence Centre for Epidemiology and Health Services Research for Healthcare Professionals (CVcare), University Medical Centre Hamburg-Eppendorf (UKE), 20246 Hamburg, Germany; 2Institute for Occupational and Maritime Medicine (ZfAM), University Medical Centre Hamburg-Eppendorf (UKE), 20246 Hamburg, Germany; 3Laboratory of Fenner and Colleagues, Bergstrasse 14, 20095 Hamburg, Germany; 4Asklepios Campus Hamburg, Semmelweis University, Lohmühlenstrasse 5, 20099 Hamburg, Germany; 5Division of Infectious Diseases, Department of Internal Medicine, University Medical Center Hamburg-Eppendorf, Martinistraße 52, 20246 Hamburg, Germany; 6German Center for Infection Research (DZIF), Partner Site Hamburg-Lübeck-Borstel-Riems, 20246 Hamburg, Germany; 7Department of Occupational Medicine, Hazardous Substances and Public Health (AGG), Institution for Statutory Accident Insurance and Prevention in the Health and Welfare Services (BGW), 22089 Hamburg, Germany

**Keywords:** SARS-CoV-2, nursing staff in outpatient care, seroprevalence

## Abstract

The risk of SARS-CoV-2 infection is particularly high for healthcare workers during the pandemic. Home care workers visit many different households per shift. Encounters with mostly elderly patients and their relatives increase the potential for the undetected spread of SARS-CoV-2. In order to gain insight into the seroprevalence of SARS-CoV-2 antibodies and possible transmission risks in outpatient care, this follow-up study was conducted with nursing services in Hamburg. The aim was to estimate the dynamics of seroprevalence in this occupational group over a 12-month period, to identify occupation-specific risk factors, and to collect information on the vaccination status of the surveyed nursing staff. Antibody testing for SARS-CoV-2 IgG against the S1 domain (EUROIMUN Analyser I^®^ Lübeck, Germany) was performed on participating healthcare workers with patient contact at a total of four time points within one year from July 2020 to October 2021 (baseline, follow-up after three, six and twelve months). The data were mostly analysed descriptively. Differences in IgG titres were analysed using variance analysis methods, particularly Tukey’s range test. The seroprevalence was 1.2% (8/678) at baseline and 1.5% (9/581) at the three-month follow-up (T1). At the second follow-up (T2) after six months, vaccination against SARS-CoV-2 was available from January 2021 onwards. The prevalence rate of positive IgG antibodies relative to the S1 domain of the spike protein test among unvaccinated individuals was 6.5%. At (T3) after twelve months (July to October 2021), 482 participants were enrolled, and 85.7% of the workers were considered fully vaccinated at this time point, while 51 individuals were unvaccinated. The prevalence was 13.7% (7/51). In our study, a low seroprevalence was found among home care workers, which was lower than in our studies conducted in the clinical setting. Therefore, it can be assumed that the occupational risk of infection is rather low for both the nursing staff and the patients/clients cared for in the outpatient setting. The good provision of protective equipment and the high vaccination rate of the staff probably had a positive influence.

## 1. Introduction

The risk of infection during the SARS-CoV-2 pandemic is particularly high for health workers as they often have close contact with infected patients [1,2]. Health insurance data in Germany show that the rate of hospitalisation due to COVID-19 among health workers was 2.4 times higher than among those in other occupations [3]. The data from accident insurance also confirm a high risk of infection for this occupational group. By May 2020, 4398 COVID-19 cases had been reported to the Institution for Statutory Accident Insurance and Prevention in the Health and Welfare Services (BGW). This figure increased to 84,728 by May 2021. The majority of cases involved inpatient care facilities and hospitals (39.5% and 37.6%, respectively). Nursing staff accounted for the largest proportion of those affected (68.8%) [4].

After acute SARS-CoV-2 infection, there is also the risk of long-term effects. In a study from Germany, health workers suspected of having an occupational COVID-19 infection were asked about their infection in 2020 and their current state of health. Half of all the participants (51%) had a SARS-CoV-2 infection in the first half of 2020. At the time of the survey, 74.2% (n = 1523) of participants reported ongoing symptoms since their infection. In terms of the duration of symptoms, it was found that 50% of those affected had been suffering from the consequences of the disease for nine months or longer (maximum of 15 months) [5].

A systematic review from 2020 included 49 studies involving a total of 127,480 health workers. The calculated seroprevalence for SARS-CoV-2 antibodies was 8.7%. However, there were regional differences, with a seroprevalence of 12.7% reported in North America and 8.5% in Europe [6]. The studies on seroprevalence focused on hospital employees. In Germany, the seroprevalence rate of health workers was rather low after the first wave of the pandemic. A study from the St. Antonius Hospital in Eschweiler from April 2020 revealed a cumulative incidence of 3.9% for a positive PCR or a positive antibody test. Here, employees in contact with COVID-19 patients had an increased risk of infection as well [7]. A study conducted in Bonn at the beginning of the pandemic (March–April 2020) found that 1% of health workers were infected [8]. By comparison, the overall infection rate among 1253 employees of the University Medical Centre Hamburg-Eppendorf in Hamburg (study period: March to July 2020) was 1.8% (n = 22). The seroprevalence among employees with direct patient contact was not significantly higher than that of other hospital staff, although more than 170 patients with SARS-CoV-2 infections were being treated at the hospital at the time [9]. However, higher infection rates were found in some instances in other studies around the world, e.g., in Belgium (12.6%) [10], Spain (11.2%) [11], Italy (14.4%) [12], and Sweden (19.1%) [13].

There are very few studies on the seroprevalence of employees in geriatric care. Nevertheless, there were repeated outbreaks of infection in nursing homes at the beginning of the pandemic. In Lower Saxony, infections had been detected in around 80 elderly care facilities by April 2020. According to the Robert Koch Institute, around one-third of all deaths from COVID-19 in the first wave occurred in retirement and nursing homes and other care facilities. According to an analysis carried out by the London School of Economics, about half of all deaths caused by COVID-19 (42% to 57%) in Italy, Spain, Ireland, Belgium, and France were reported from nursing care facilities. The elderly and those in need of nursing care therefore require particular protection. Due to the immune system weakening with age, the risk of infection is higher, and the course of the disease is often more severe. The presence of pre-existing illnesses, such as type 2 diabetes mellitus or cardiovascular diseases and cancer, and factors such as obesity and smoking increase the risk of developing severe COVID-19. An increased mortality rate was observed in the elderly and in those with chronic diseases [14].

Nursing staff working in outpatient care visit many different households per shift. These encounters, which involve mostly elderly patients and their relatives, increase the potential for the SARS-CoV-2 virus to spread undetected. To gain insights into seroprevalence and possible transmission routes in outpatient nursing care, a follow-up study of nursing services in Hamburg was conducted as part of the National Research Network of University Hospitals for the scientific monitoring of the COVID-19 pandemic. The prevalence of specific SARS-CoV-2 antibody titres was investigated using blood sampling. The objective was to estimate the dynamics of seroprevalence in this occupational group over 12 months, to identify occupation-specific risk factors, and to collect information on the vaccination status of the surveyed nursing staff.

## 2. Materials and Methods

### 2.1. Study Design

Employees in outpatient care services in Hamburg were included in this observational study (July 2020–October 2021). Antibody tests for SARS-CoV-2 IgG (EUROIMUNN Analyser I^®^) were performed on the participating nursing staff over a total of four sessions within a twelve-month period (baseline and follow-up after three, six, and twelve months).

Nursing care services in Hamburg were informed about the study via a letter to their managers. Only outpatient care services based in Hamburg were able to participate, and inpatient facilities were not included. After actively expressing their interest to the study centre by telephone or in writing, respective outpatient care services were included in the study regardless of the number of employees they have and the number of those in their care. The staff was provided with information (flyer, participant information, and hotline) about the objective and procedure of the study. Interested employees within the care services were able to sign up for an appointment for blood sampling with the care service managers or directly with the study centre. Each participant received a questionnaire in advance, which collected information on socio-demographics and possible routes of infection at the workplace and in the individuals’ private environment, as well as symptoms and stress. The study periods each lasted about twelve weeks, as individual care services were included in stages for logistical reasons. 

A mobile team took the blood samples on the premises of the care services. If individual employees were unable to attend these appointments, substitute- appointments were offered at the participating institutes. Each subsequent test appointment involved a questionnaire focused on the current infection status, possible symptoms, and willingness to be vaccinated. The baseline examinations were conducted between July and September 2020. T0 after 3 months was carried out between October 2020 and December 2020. 

From the T2 test point after six months (January to March 2021), different options for vaccinating against SARS-CoV-2 were available, so the individuals’ current vaccination status was also recorded from this point onwards. 

At the last test point (T3), from July up to and including October 2021, all employees with a complete vaccination status (1 time with Johnson & Johnson or 2 times with mRNA vaccines or vector vaccines) were offered the additional option of having their blood samples analysed for antibodies against the N protein, neutralising antibodies and the T-cell response. Additional written consent was obtained from employees for these tests, which were a new addition to the study’s protocol. All participants were informed in a written report about their test results at each time point, and an explanation of how these results should be interpreted was provided.

### 2.2. Serological Methods

The most noteworthy results are the antibody test results against SARS-CoV-2. SARS-CoV-2 serology was performed on the participants at all four points in the study. All tests were carried out according to the manufacturer’s specifications. The blood samples were tested for SARS-CoV-2 IgG using an enzyme-linked immunosorbent assay (ELISA) in an accredited laboratory (EUROIMMUN^®^, Lübeck, Germany). In the case of a positive test result, the findings were verified using the Roche^®^ Elecsys SARS-CoV-2 assay. SARS-CoV2 IgG was tested against the S1 domain of the spike protein in an accredited laboratory using an enzyme-liked immunosorbent assay (ELISA) (EUROIMMUN, Lübeck, Germany). IgG against the spike protein was detected both after vaccination and infection. However, the vaccines used in the European Union during study time stimulated only the production of IgG against the spike protein. Therefore, in the case of a positive result, IgG against the nucleocapsid-protein (NCP) was measured using an ELISA on the EUROIMMUN Analyzer I (EUROIMMUN, Lübeck, Germany) to detect a precedent infection. 

As evidence of antibodies against the spike protein, the values of results that were more than 1.1 were considered positive, while values from 0.8 to 1.1 results were considered as borderline cases. A positive test result indicates the formation of antibodies against the spike protein. Antibody formation is due to vaccination, infection, or a combination of both. If the result was positive in both tests, we assumed that an infection had occurred, and the person tested was recorded as positive provided that no vaccination had been carried out. The participants were informed of the test result in writing in each case. 

#### Additional Antibody Tests at T3 

Neutralising antibodies relative to SARS-CoV-2 were measured using the TrimericS IgG^®^ assay (DiaSorin, Saluggia, Italy), which detects neutralising antibodies relative to RBD and NTD epitopes. Values above 33.8 BAU/mL are considered positive as evidence of neutralising antibodies, while values ≤33.8 BAU/mL are considered negative. A positive result indicates the formation of neutralising antibodies that block the uptake of the virus into the body’s own cells. The detection of antibodies indicates an immune response following vaccination, infection, or a combination of both. 

For the detection of antibodies against the IgG nucleocapsid protein, values of more than 1.1 are considered positive, while values from 0.8 to 1.1 are considered as borderline cases. Positive or borderline values are consistent with antibody formation following a SARS-CoV-2 infection. Unlike IgG antibodies, IgM antibodies are formed immediately after initial contact with the pathogen. An elevated IgM value therefore signals an acute infection with a pathogen. However, IgM antibodies are formed a few days earlier than IgG antibodies, and unlike IgG antibodies, they decrease again after four to five weeks [15].

SARS-CoV-2 T-cell immunity was determined using the interferon-gamma release assay (Quan-T-Cell^®^, EUROIMMUN, Lübeck, Germany), which measures interferon-gamma levels stimulated by CD4+ and CD8+ T-cells via the spike protein. Results below 100 mIU/mL were negative, those from 100 to 200 mIU/mL were borderline, and those above 200 mIU/mL were positive. A positive test result means that T lymphocytes directed against the SARS-CoV-2 were generated as a result of vaccination, infection, or a combination of both.

### 2.3. Data Management and Statistical Analysis

The data were collected in pseudonymised form using an individual participant code. Address information was only used to make test appointments and for sending test results. The encrypted identification list used for this purpose was stored electronically at the study centre. Only authorised persons who were not involved in the analysis had access to this. The encryption lists were deleted after the completion of the study, and the signed informed consent forms will be archived separately from the test results for ten years. All pseudonymised data will also be kept for ten years in accordance with regulations. The serum samples will be destroyed after no more than two years. All data protection requirements were complied with. Every participant agreed to participate in writing. The Ethics Committee of the Hamburg Medical Association (PV7298) was consulted. 

The data were analysed descriptively for the most part. We indicated missing values in the tables. Metric variables were described in terms of the arithmetic mean, median, standard deviation (SD), or standard error (SE). We analysed differences in IgG titres using variance analysis methods, particularly Tukey’s range test. We used SPSS software (version 27, SPSS Inc., Armonk, NY, USA) to evaluate the data. R (version 4.2.1, R Foundation for Statistical Computing, Vienna, Austria) was used to visualise the results [16].

## 3. Results

### 3.1. Description of the Study Population

For the baseline examination conducted in July 2020, a total of 678 study participants from a total of 51 outpatient care services in Hamburg were included. In total, 81% of the participants were female, and the mean age was 46.4 (SD 12.1) years. Nearly 66% of the employees specified their work as caring for clients. This was followed by household assistance activities such as cleaning and shopping (22.4%) (Table 1). The most frequently stated occupation was trained geriatric nurses (27.3%). Possible risk factors for seroconversion, such as a pre-existing illness or contact with infected colleagues or clients, are shown in Table 2. At the time of the baseline study, 92% of the respondents stated that sufficient protective equipment had been available. Only two employees reported having had contact with an infected client. However, no contact took place without protection (Table 2).

The results of the antibody tests throughout the study period are shown in Figure 1. Eight employees had a positive antibody test at baseline. This corresponds to a seroprevalence of 1.2%. At follow-up after three months (T1), 581 employees were tested. An additional positive finding increased the seroprevalence to 1.5%. The increase in prevalence during this period was 0.3%. By the second follow-up (T2) after six months, from January 2021, vaccination against the SARS-CoV-2 virus was available, and the IgG test produced a positive result in 36.4% (176/483) of those tested. A total of 261 (59.6%) employees reported that they had not yet received a vaccination during this study period. The prevalence rate for a positive IgG test among the unvaccinated was 6.5%. 

At the last test point (T3) after twelve months (July to October 2021), 482 participants were included. A total of 462 employees provided information on their vaccination status. In total, 396 (85.7%) employees were considered fully vaccinated at this time, and the percentage of positive IgG test results among those vaccinated was 98.5%. Moreover, 51 individuals were not vaccinated at this point, and the prevalence rate among them was 13.7% (7/51). 

Data relating to different antibody tests were available for 325 vaccinated employees. A total of ten (3.1%) employees were found to have antibodies relative to both the spike and N proteins, suggesting that infection with SARS-CoV-2 had also occurred. One non-responder was observed, which means that no IgG antibodies or T-cells response were detectable, and we also did not detect a T-cell response in 32 (9.8%) employees (Figure 1). 

### 3.2. Vaccination Status and IgG Titres (Spike Protein)

At the final test point in the study, we asked employees who were considered fully vaccinated about the vaccine combinations they had received. There were a total of six different combinations that were considered complete vaccinations at the corresponding time. Employees most frequently received two doses of the BioNTech/Pfizer vaccine (n = 193), followed by individuals vaccinated with a combination of AstraZeneca and BioNTech/Pfizer vaccines (n = 31) (Figure 2). The highest median IgG titres against the SARS-CoV-2 spike protein were observed with combinations of the AstraZeneca vaccines with a product from BioNTech/Pfizer (n = 31) or Moderna (n = 16) and the combination of two vaccinations with the Moderna vaccine (n = 12). Statistically speaking, significantly lower IgG titres were found for the combination of the double vaccination with the AstraZeneca vaccine (n = 15) and the single vaccination with the Johnson & Johnson vaccine (n = 6), which is also considered a complete vaccination.

## 4. Discussion

This is the first longitudinal study of work-related SARS-CoV-2 infections among employees in outpatient care services in Germany. During the period of the first two surveys between July 2020 and the end of December 2020, we found a low seroprevalence of 1.2% and 1.5%, respectively. The increase in prevalence during this period was 0.3%. At the second follow-up from January 2021, the vaccinations led to an increase in antibody titres (IgG) against the spike protein; among the unvaccinated, seroprevalence increased from 6.5% to 13.7% over the course of the year. It is difficult to assess whether this indicates an actual increase in infections or whether only the denominator shrinks, as the group of unvaccinated individuals must be regarded critically. There may have been infected people who had to postpone vaccination because of the infection. At the last examination point in our study, several of the 51 unvaccinated participants reported that they had already tested positive for COVID-19. Therefore, they may not have been eligible for vaccination up to that point. On the other hand, unvaccinated individuals had a higher risk of infection than those who were vaccinated over the course of the pandemic, especially since the risk of infection in Germany was significantly higher in the second wave of the pandemic (from October 2020 to February 2021). This is probably due to a mixed effect, as the prevalence rate quadrupled from 1.5% to 6.6%, but the absolute number of affected persons only approximately doubled from 9 to 16. 

After twelve months, in 325 subjects with complete vaccination protection, it was also possible to distinguish whether the seropositivity was due to past infection or vaccination. Only ten (3.1%) subjects had antibodies against the spike and nucleocapsid proteins, which indicates a past infection. It can therefore be assumed that there was a low risk of infection for the employees in Hamburg during the twelve-month study period. The provision of protective equipment such as face masks, gloves, and protective gowns was rated as very good by the employees and offered protection against infection. This, in addition to the special situation of providing outpatient care in the household environment, may have had an influence on the low seroprevalence. Unlike in clinical or inpatient settings, there was very little contact with infected clients or colleagues in our study, and there was no evidence of SARS-CoV-2 outbreaks such as those that occurred in inpatient geriatric care facilities. 

A comparison with other studies in outpatient settings is only possible to a limited extent due to a lack of studies on this occupational group. Although there are numerous studies on health workers worldwide, the focus has mostly been on employees in hospitals who frequently come into contact with infected patients [17,18,19,20,21,22]. 

The Swiss SEROCoV-WORK+ study assessed seroprevalence among employees recruited after the first wave of the COVID-19 pandemic in Geneva, Switzerland. Employees from 16 sectors and 32 different occupations were tested for anti-SARS-CoV-2 IgG antibodies (18 May to 18 September 2020). Out of 10,513 participants, 1026 (9.8%) were seropositive. The rates ranged from 4.2% in the media sector to 14.3% in the nursing home sector. However, considerable differences were observed within the individual domains: nursing homes (0.0–31.4%), home-based care (3.9–12.6%), healthcare (0.0–23.5%), public administration (2.6–24.6%), and public safety (0–16.7%) [17].

When comparing the seroprevalence of our study with the results from seroprevalence studies among hospital staff, the rates in Germany often turn out to be somewhat higher.

Results from an initial Germany-wide population-based seroepidemiological study conducted in the 1st 2 waves of the pandemic with 7879 subjects, 915 of whom were health workers, showed that working in the healthcare sector was associated with roughly twice the risk (OR 2.1) of contracting a SARS-CoV-2 infection when potential risk factors were taken into account. The overall prevalence among employees in healthcare professions was 4.6% compared to other employees at 1.8% [18].

Herzberg et al. [19] investigated employees in a hospital in Northern Germany in a prospective longitudinal study carried out between March and June 2020. The aim of this study was to detect the SARS-CoV-2 virus and specific antibodies (IgG) among the employees. Anti-SARS-CoV-2 IgG antibodies were found in 38 out of 871 (4.4%) study participants. The authors concluded that the low prevalence rate may be indicative of good hygiene management. 

Following the first wave of infection in the summer of 2020, the Rhein-Maas Hospital (RMK) in Würselen offered all staff the opportunity to be tested for SARS-CoV-2 by means of antibody testing. Occupational and non-occupational risk factors for infection were surveyed. In total, 903 employees took part in this cross-sectional study; 5.8% of the employees had a positive PCR test result in their medical history or tested positive in the IgG test. There were differences relating to areas of work, with employees in at-risk jobs showing an increased OR of 1.9 (95% CI 1.04–3.5). An increase in occupational infection risk was found even after controlling for non-occupational infection risks [20].

At a hospital in North Rhine-Westphalia, employees were tested four times within a year using PCR and serology. Swab and blood tests were carried out simultaneously between April 2020 and April 2021. The study included 1506 health workers, with a cumulative incidence after one year of 10.6% (165/1506). Working in an intensive care unit or onwards with patient contact was found to be a risk factor (OR 4.4; 95% CI 1.73–13.6 and OR 2.9; 95% CI 1.27–8.49) for a positive test result [21]. 

Tomcyk et al. carried out a longitudinal study in 2020 in a Berlin tertiary hospital [22], where the employees were tested by means of blood samples in May/June and December 2020. In May/June, 18/1477 (1.2%) employees were seropositive for SARS-CoV-2, followed by 56/1223 (4.6%) in December. Of those tested in May/June, all those who were ELISA seropositive were still seropositive after six months.

In contrast, there was no relevant increase in seroconversions due to previous infections in our study. Although the IgG seroprevalence among the unvaccinated increased to 13.7% over the course of the year, at the end of the twelve-month period, only ten (3.1%) subjects showed evidence of a past infection (antibodies to the N protein), which, on the whole, indicates a low occupational risk of infection in outpatient care. However, it is important to note that the antibodies relative to the nucleocapsid protein may have disappeared over time in some subjects. 

### Limitations and Strengths

Our study has several limitations that should be taken into account. Participation in the study was voluntary, and the participating institutions were not randomly selected, which could lead to bias in the sample. The data were collected under real-life conditions. Tests and data collection using questionnaires had to be organised in such a way as to avoid making what was already a stressful work situation even more difficult as a result of the study. Possible exposure to occupational infection risk was therefore only recorded in detail at the baseline. Our study is a longitudinal study conducted over a period of twelve months. This led to changes in the general conditions in response to the dynamic pandemic setting. For one thing, vaccination against SARS-CoV-2 was available after six months, and for another, SARS-CoV-2 mutated with various waves of the pandemic from wild type to delta variants and further to the omicron variant, with their corresponding differences in virulence. This may have resulted in a different exposure risk among employees.

The dynamic nature of the pandemic also makes comparison with other studies difficult, as the test points vary from study to study.

Due to organisational reasons, the test sessions took place at intervals, making it impossible to determine point prevalence at specific points in time in our study. 

Nevertheless, our study is one of the few conducted on outpatient care workers who perform an important task in the care of elderly and vulnerable people in Germany. The study thus contributes towards assessing the risk of infection for both the nursing staff and the people in need of care and towards evaluating the existing protective measures in place.

## 5. Conclusions

In our study, we found a low seroprevalence among outpatient care workers. It was lower than the seroprevalence in our conducted studies in clinical settings. It is therefore plausible that the occupational risk of infection is relatively low for both the employees themselves and the persons receiving care in outpatient settings. Factors that probably had a positive influence in this regard were the adequate supply of protective equipment and the high vaccination rate among employees. 

## Figures and Tables

**Figure 1 ijerph-20-05599-f001:**
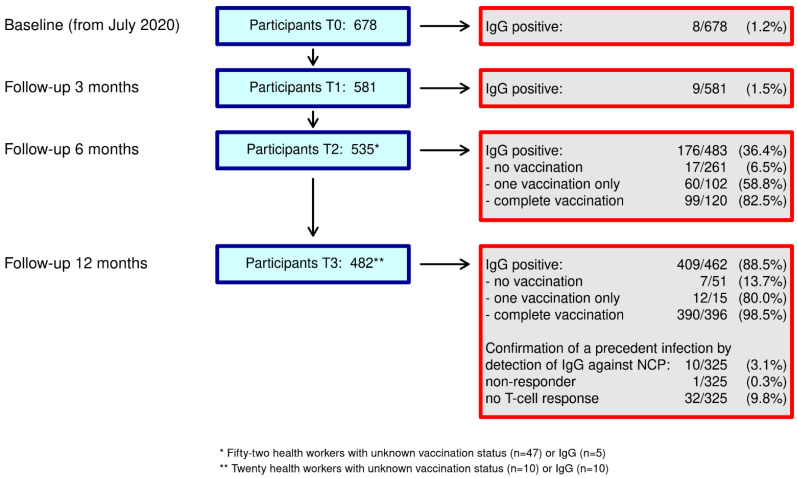
Flow chart of the number of participants involved in IgG seroprevalence investigations among outpatient care staff in Hamburg, longitudinally, over the study period from July 2020 to October 2021.

**Figure 2 ijerph-20-05599-f002:**
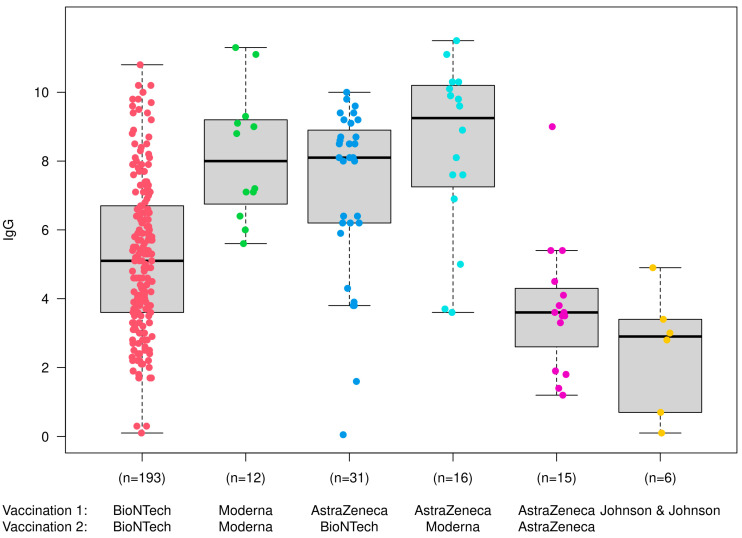
Boxplot of the IgG titres of fully vaccinated study participants at point T3, differentiated by vaccine combination (n = 273).

**Table 1 ijerph-20-05599-t001:** Description of the study population (n = 678).

Variable	Missing Values	Quantity/Average	Percentage/Standard Deviation
Gender	3		0.4%
female		549	81.0%
male		126	18.6%
non-binary		0	0.0%
Age (years)	6	46.4	12.1
Field of work	10		1.5%
outpatient care at home		446	65.8%
assistance (shopping, etc.) at home		152	22.4%
assistance for independent living		45	6.6%
office work, care service administration		125	18.4%
management		25	3.7%
other		47	6.9%
Training (care-related)	47		6.9%
nurse		131	19.3%
geriatric nurse		185	27.3%
nursing assistant		67	9.9%
remedial therapist		3	0.4%
social education assistant		11	1.6%
trainee nurse		16	2.4%
no care-related training		127	18.7%
other		142	20.9%
Type of client care	51		7.5%
Basic care		392	57.8%
Treatment care		335	49.4%
Intensive care		59	8.7%
no care activities		226	33.3%

**Table 2 ijerph-20-05599-t002:** Possible risk factors for a SARS-CoV-2 infection.

Belonging to a COVID-19 Risk Group?	31		4.6%
no		457	67.4%
yes		190	28.0%
Risk disease: diabetes		27	4.0%
Risk disease: high blood pressure		97	14.3%
Risk disease: asthma		59	8.7%
Risk disease: other		68	10.0%
Number of adults in own household	10		1.5%
no other adults		173	25.5%
1 other adult		370	54.6%
2 or more other adults		125	18.4%
Number of children in own household	89		13.1%
no children		318	46.9%
1 child		137	20.2%
2 or more children		134	19.8%
Use of public transport	9		1.3%
not at all		361	53.2%
occasionally		191	28.2%
daily		117	17.3%
Is sufficient protective equipment available?	27		4.0%
no		28	4.1%
yes		623	91.9%
Protection: face mask	0	580	85.5%
Protection: FFP mask	0	145	21.4%
Protection: disposable gloves	0	508	74.9%
Protection: protective gown	0	89	13.1%
Protection: protective goggles	0	26	3.8%
Travel abroad within the past four weeks	9		1.3%
no		619	91.3%
yes		50	7.4%
Direct contact with a person who was found to have SARS-CoV-2?	11		1.6%
no		661	97.5%
contact in the private sphere		1	0.1%
contact with infected work colleague		1	0.1%
contact with infected patients without protective clothing		0	0.0%
contact with infected patients with protective clothing		4	0.6%

## Data Availability

The data are available upon request from the corresponding author.

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
