# Peer review of "Longitudinal SARS-CoV-2 Seroprevalence among Employees in Outpatient Care Services in Hamburg"

_ijerph, 2023, doi:10.3390/ijerph20085599_

Round 1
Reviewer 1 Report
The manuscript studied seroprevalence among long term care faciltiy staffs in Hamburg. This is fairly intresting to readers for the journal.
There are no major comments but a few minor comments;
1. Enrollment of participants is not clear. Do authors provide any feedback of results or insentives to participant? It may cause bias in enrollment.
2. How did authors manage seropositive cases of T1 in T2 and T3? In figure1, I could not see a category of previous infection.
3. How long do authors collect blood samples from participants in each tier? I dont know COVID19 situation in Hamburg while taking a month or more may change infection background of participants.
4. In page 9, may remove h before the employees.
Author Response
Answer to the Reviewer
There are no major comments but a few minor comments;
1. Enrollment of participants is not clear. Do authors provide any feedback of results or insentives to participant? It may cause bias in enrollment.
Answer: Thank you or this comment. All participants were informed in a written report about their test results at each time point and explained how these results should be interpreted. See methods section. There were no incentive or incentive payments. Nevertheless, a selection bias cannot be ruled out and is addressed under Limitations.
How did authors manage seropositive cases of T1 in T2 and T3? In figure1, I could not see a category of previous infection.
Answer: That's right, we haven't created any categories for this. At the time of the blood samples, all subjects were healthy and able to work. Therefore, no isolation measures applied. All participants were testet again. So some IG values decreased during the time slots. In addition, at T1, T2 and T3, many had already been vaccinated, so that many positive test results were also due to vaccination. Only at T3 was it possible to distinguish whether the test was positive due to a previous infection or vaccination. Therefore, the statement here can only be interpreted to a very limited extent. No categorisation was made.
How long do authors collect blood samples from participants in each tier? I dont know COVID19 situation in Hamburg while taking a month or more may change infection background of participants.
Answer to the Reviewer: Thanks for your comment. One or two appointments for blood collection were made at each facility. Because 51 different care services were involved and because the procedure for blood collection was very complex, we needed up to 3 months during the individual study points until all blood tests were completed. In Hamburg, the infection dynamics were not so significant that this resulted in major differences in infection risk.
In page 9, may remove h before the employees.
Answer: Thanks we removed h.

Reviewer 2 Report
The paper describes a study on SARS-CoV-2 seroprevalence over 12 months (4 time points, of about 12 weeks each) of 678 employees in outpatients care services in Hamburg.
- in the second to the last sentence of the abstract, it is stated that "risk of infectione is rather low for both the employees and the nursing staff", but I did not find specific seroprevalence values for the nursing staff and other job categories; please clarify.
- in the abstract, line 24, it seems that antibody test was performed only in nurses, please clarify if all the emplyees were actually nurses.
- on pages 6 and 8 , the "incidence" seems the difference between the two prevalences, you may consider to speak of a prevalence that increased of 0.3 percentage point only.
Author Response
Answer to the Reviewer 2
The paper describes a study on SARS-CoV-2 seroprevalence over 12 months (4 time points, of about 12 weeks each) of 678 employees in outpatients care services in Hamburg.
- in the second to the last sentence of the abstract, it is stated that "risk of infectione is rather low for both the employees and the nursing staff", but I did not find specific seroprevalence values for the nursing staff and other job categories; please clarify.
Answer: Thanks. Sorry we changed this correct in Nursing staff and patients/clients to be cared for
- in the abstract, line 24, it seems that antibody test was performed only in nurses, please clarify if all the emplyees were actually nurses.
Answer: Thank you for pointing this out. All workers with contact to their clients/patients could participate. The occupations and professions of the test persons are listed in Table 1. However, the patients were excluded because it would have been too much work. We have changed this accordingly.
- on pages 6 and 8 , the "incidence" seems the difference between the two prevalences, you may consider to speak of a prevalence that increased of 0.3 percentage point only.
Answer: Thank you we changed this accordingly.
